# The Performance of Cellulose Composite Membranes and Their Application in Drinking Water Treatment

**DOI:** 10.3390/polym16020285

**Published:** 2024-01-20

**Authors:** Rengui Weng, Guohong Chen, Xin He, Jie Qin, Shuo Dong, Junjiang Bai, Shaojie Li, Shikang Zhao

**Affiliations:** 1Institute of Biology and Chemistry, Fujian University of Technology, Fuzhou 350118, China; 2College of Ecological Environment and Urban Construction, Fujian University of Technology, Fuzhou 350118, China; 13052696442@163.com (G.C.); baijunjiang0511@163.com (J.B.);

**Keywords:** cellulose, drinking water, ZrO_2_, nanofiltration membrane, ultrafiltration membrane, separation performance

## Abstract

Water scarcity and water pollution have become increasingly severe, and therefore, the purification of water resources has recently garnered increasing attention. Given its position as a major water resource, the efficient purification of drinking water is of crucial importance. In this study, we adopted a phase transition method to prepare ZrO_2_/BCM (bamboo cellulose membranes), after which we developed IP-ZrO_2_/BC-NFM (bamboo cellulose nanofiltration membranes) through interfacial polymerization using piperazine (PIP) and tricarbonyl chloride (TMC). Subsequently, we integrated these two membranes to create a combined “ultrafiltration + nanofiltration” membrane process for the treatment of drinking water. The membrane combination process was conducted at 25 °C, with ultrafiltration at 0.1 MPa and nanofiltration at 0.5 MPa. This membrane combination, featuring “ultrafiltration + nanofiltration,” had a significant impact on reducing turbidity, consistently maintaining the post-filtration turbidity of drinking water at or below 0.1 NTU. Furthermore, the removal rates for COD_MN_ and ammonia nitrogen reached 75% and 88.6%, respectively, aligning with the standards for high-quality drinking water. In a continuous 3 h experiment, the nanofiltration unit exhibited consistent retention rates for Na_2_SO_4_ and bovine serum protein (BSA), with variations of less than 5%, indicating exceptional separation performance. After 9 h of operation, the water flux of the nanofiltration unit began to stabilize, with a decrease rate of approximately 25%, demonstrating that the “ultrafiltration + nanofiltration” membrane combination can maintain consistent performance during extended use. In conclusion, the “ultrafiltration + nanofiltration” membrane combination exhibited remarkable performance in the treatment of drinking water, offering a viable solution to address issues related to water scarcity and water pollution.

## 1. Introduction

The demand for freshwater resources has grown substantially in the 21st century as society continues to advance [1]. At the same time, public awareness of environmental issues has gradually improved [2]. Among these water-related issues, the safety of drinking water, which directly impacts human health, has garnered increased attention [3,4,5]. Therefore, the search for effective methods to remove low-concentration pollutants in drinking water without using chemicals or producing toxic by-products has become a research hotspot [6,7,8,9,10,11,12].

Membrane separation technology underwent global development between the mid-18th century, when the concept of membrane separation was introduced, and the 1950s. In the 1960s, the emergence of asymmetric membranes in separation technology facilitated the industrialization of membrane separation and broadened its applications. Jihua Hao et al. introduced the technology of membrane treatment for wastewater treatment in industry and made suggestions for future development [13]. Stern, S. A. et al. studied the application of membrane separation technology to a gas separation field and discussed the effects of several important process variables on the single-stage separation, and separation membrane area requirements are outlined in a parametric study [14]. Bin Liang et al. provide an overview of the most crucial organic mixtures requiring separation, the primary separation processes currently employed in organic solvents, and the recent advancements in newly developed membranes [15]. C. Visvanathan et al. summarized the research work on membrane separation bioreactors for wastewater treatment and reviewed how and why it was developed and applied [16]. By the end of the 20th century, various membrane types, including ultrafiltration, nanofiltration, and even reverse osmosis, had been successfully developed, thus contributing to the maturation of membrane separation into a full-blown industrial technology. Wang H et al. introduced the working principles, features, and classifications of membrane technology, offering insights into the anticipated trends in its application development [17]. Carolina Fonseca Couto et al. presented the application of membrane technology for removing novel pharmaceutical pollutants from water, with a specific focus on pharmaceutically active compounds [18]. G. Ciardelli et al. investigated the application of membrane separation technology in the industrial recovery of dyeing and weaving wastewater, demonstrating the economic feasibility of this method [19]. MaryTheresa M. Pendergast et al. conducted a study on the application of membrane technology in water treatment, emphasizing the successful application and widespread adoption of nano-mixed membranes in commercial settings [20].

Currently, membrane technologies for drinking water treatment mainly include microfiltration, ultrafiltration, nanofiltration, and reverse osmosis [21,22], with ultrafiltration and nanofiltration being the most widely used. Microfiltration and ultrafiltration effectively remove pollutants such as suspended solids, bacteria, and proteins, thereby improving the quality of drinking water by maintaining low turbidity [23,24]. Ultrafiltration membrane devices are small, easy to replace, and commonly employed in daily front-end treatment. However, due to their material characteristics, ultrafiltration membranes cannot efficiently remove certain pollutants. Given their smaller pore sizes, nanofiltration membranes intercept pollutants with lower molecular weight. The “Donnan effect” resulting from their negative surface charge aids in removing organic matter and ions, including natural organic matter and disinfection by-products in water [25]. Additionally, the removal of inorganic ions enhances the taste of drinking water [26].

Both ultrafiltration and nanofiltration membranes have their own limitations. Among them, the stability of the membrane is one of the key problems in membrane technology [27]. During the long-term operation of the membrane, suspended matter or organic matter in the water can cause serious membrane pollution and reduce the membrane flux, leading to unstable water production [26]. The combined membrane treatment technology of ultrafiltration and nanofiltration has become an increasingly popular research area for drinking water treatment [28,29]. Using ultrafiltration as a pre-treatment process prior to nanofiltration can effectively remove suspended matter and some organic matter in raw water, thereby reducing membrane pollution in the subsequent nanofiltration step and improving the service life of the membrane [30]. In the nanofiltration process, organic matter, ions, and other pollutants in the ultrafiltration-produced water are removed, complementing each other to improve the quality of drinking water.

Bomou Ma et al. successfully fabricated a cellulose hollow fiber membrane with relatively high tensile strength and rejection rate; however, it exhibited limited water flux. Milad Rabbani Esfahani et al. developed a bamboo cellulose membrane with excellent rejection rate and water flux; nevertheless, compared to other ultrafiltration membranes, this new cellulose membrane exhibited weaker fouling resistance capability [31]. Li et al. prepared a cellulose-based nanofiltration membrane (LBL-NF-CS/BCM). The authors reported that the LBL-NF-CS/BCM composite membranes achieved a rejection rate of approximately 36.11% for a 500 ppm NaCl solution under the given conditions while maintaining a membrane flux of approximately 12.08 L/(m^2^·h) [32]. Afterward, Shi Li et al. employed an interfacial polymerization technique to create a hydrophilic bamboo cellulose nanofiltration membrane (IP-NF-BCM) [32]. This membrane exhibited a notable rejection rate of 40% for NaCl, coupled with a water flux of 15.64 L/(m^2^·h). Weng et al. successfully synthesized a novel cellulose nanofiltration membrane using the phase inversion method. The membrane displayed excellent stability during water treatment processes and proved effective in removing organic compounds from aqueous solutions [33,34,35].

Although researchers continue to improve the performance of ultrafiltration or nanofiltration membranes, there are still many inevitable shortcomings that must be addressed [36]. For example, the membrane holes are easily clogged and the membrane is easily contaminated and denatured [37]. Therefore, using a combined membrane process consisting of ultrafiltration and nanofiltration membranes can greatly enhance water purification performance [38,39,40]. The use of ultrafiltration membranes as a pretreatment process for suspended solids and some organics in the raw water before nanofiltration can alleviate the membrane contamination problem in the next step of the nanofiltration unit and improve the service life of the membranes. After ultrafiltration, the nanofiltration process removes organic matter, ions, and other pollutants from the ultrafiltration produced water. Ultrafiltration and nanofiltration complement each other to improve drinking water quality.

In this experiment, advanced drinking water treatment was achieved through a combined “ultrafiltration + nanofiltration” cellulose membrane assembly process involving ZrO_2_/BCM and IP-ZrO_2_/BC-NFM. Common indicators of filtered water quality, such as turbidity, COD_MN_, ammonia nitrogen, and total hardness, were assessed in this experiment, and the treatment effect was evaluated according to the “Drinking Water Sanitation Standard (GB5749-2022) [41].” Additionally, by simulating pollutants, the interception effect and flux change of each unit in the cellulose membrane assembly were detected. The stability of the membrane was then evaluated by testing the flux recovery rate of the membrane.

## 2. Materials and Methods

### 2.1. Materials and Instruments

All chemicals used in this study were acquired from Shanghai Aladdin Biochemical Technology (Shanghai, China) Co., Ltd. The membrane separation system (model number KCT45-70) was obtained from Xiamen Kaichengtong Machinery Equipment (Xiamen, China) Co., Ltd. Turbidity measurements were conducted using a portable turbidity tester purchased from Bell Analytical Instruments (Dalian, China) Co., Ltd. (model number BSC5300).

The membrane was fabricated using a phase inversion method, which involved blending nano ZrO_2_ with natural bamboo cellulose (BC) to create a fouling-resistant and hydrophilic cellulose ultrafiltration membrane (ZrO_2_/BCM). The ratio of cellulose, N-Methylmorpholine N-oxide (NMMO), and water was 1:8:n, and the added ZrO_2_ was 1.0wt.%. A specific quantity of ZrO_2_ nano-metal particles was introduced into the prepared NMMO aqueous solution, and ultrasonic dispersion was employed for 30 min to ensure the uniform dispersion of the nano-particles in the NMMO solution. Subsequently, antioxidants and cellulose were added to the suspension in a predetermined ratio and mechanically stirred for 2–3 h, ensuring thorough mixing and dissolution. The resulting mixture was then defrosted at 90 °C for 4–6 h to achieve a homogeneous casting film solution. The homogeneous casting solution was carefully guided fabric positioned on the coating machine, adjusting the touch, and heating the scraper onto a non-woven to 60–90 °C. The film was scraped at a controlled speed of 20 cm/min. Following the scraping process, the film was allowed to air-dry for 10–15 s before being immersed in deionized water for 24–48 h. Subsequently, the film was removed and placed indoors for natural drying, ultimately yielding the ZrO_2_/BCM membrane. Afterward, the ZrO_2_/BCM membrane served as the basis for the preparation of a nanofiltration membrane. An interfacial polymerization process was then employed, utilizing anhydrous pyridinium imidazole (PIP) as the aqueous monomer trimellitic chloride (TMC) and n-hexane as the organic phase. Through the reaction between the PIP monomer and acyl chloride monomer, a cross-linked and dense polyamide active layer was formed on the ZrO_2_/BCM substrate, resulting in the development of a novel cellulose nanofiltration membrane (IP-ZrO_2_/BC-NFM) [42,43].

In this study, the ZrO_2_/BCM and IP-ZrO_2_/BCM-NFM membrane processes were used in combination with “ultrafiltration + nanofiltration” cellulose membrane to achieve deep treatment of raw water, as shown in Figure 1. A 0.45 μm organic microfiltration membrane was used at the front end of the cellulose membrane assembly to remove suspended substances and impurities. The ZrO_2_/BCM was then used as the ultrafiltration unit for pre-treatment prior to nanofiltration, followed by IP-ZrO_2_/BCM-NFM as the main membrane process for water sample purification. The water quality indicators of the treated water samples, including chroma, turbidity, COD_MN_ (mg/L), pH, ammonia nitrogen (AN) (mg/L), and total hardness (TH) (mg/L), were tested and compared with the ultrafiltration unit. The performance of the ultrafiltration + nanofiltration combined membrane treatment process for drinking water treatment was then evaluated.

### 2.2. Measurements of Water Quality Parameters

#### 2.2.1. Turbidity

To measure turbidity, a 10-milliliter water sample was collected in a test tube. Turbidity values for both raw water and treated water were determined using a portable turbidimeter. To ensure data reliability and minimize the influence of external factors, each sample was tested three times. According to the testing criteria outlined in the “Sanitary Standards for Drinking Water” (GB5749-2022), turbidity should remain below 1 NTU (Nephelometric Turbidity Units). Therefore, reducing the turbidity of drinking water is of paramount importance to ensure water quality.

#### 2.2.2. Total Hardness

Chromium black T was used as an indicator for the determination of total hardness. Initially, the indicator was introduced into the water sample to be analyzed, and the mixture was stirred until a wine-red color emerged. Afterward, EDTA was gradually added with continuous agitation until the solution’s color transitioned from wine-red to pure blue, indicating the finalization of the titration process. The volume of EDTA used for titration was then employed to calculate the total hardness in the water sample. According to the “Sanitary Standards for Drinking Water” (GB5749-2022), the total hardness should not exceed 450 mg/L.

#### 2.2.3. Organic Matter

Chemical oxygen demand (COD) is a common parameter used to quantify the organic matter content in water. In this study, potassium permanganate was employed as an oxidizing agent to measure COD_MN_ (chemical oxygen demand for manganese) in water samples. As per the “Drinking Water Health Standards” (GB5749-2022), the COD level should not exceed 3 mg/L.

#### 2.2.4. Membrane Fouling and Stability

A solution containing Na_2_SO_4_/BSA with a concentration of 1 g/L was added to the feed in the same proportion. The solution was then passed through the membrane at 25 °C and 0.5 MPa for a total of 15 h. The membrane’s permeation fluxes were recorded both at the beginning and after filtration. Prior to the extended dynamic experiment, the membrane was pre-conditioned in the membrane filtration system for 0.5 h. Each test was conducted three times to ensure data accuracy and reliability while preventing pollution due to changes in the feed solution and other factors that might affect membrane performance. The average value, denoted as J1, was then calculated.

The membrane flux recovery rate (*r*), used to assess the membrane’s anti-pollution performance, was determined by calculating the ratio of the membrane flux to the initial membrane flux (*J_0_*). The calculation formula is as follows:(1)$$r=\frac{J_{1}}{J_{0}}\times x\times 100\%,$$

#### 2.2.5. Membrane Cleaning and Flux Recovery Rate

The membrane was treated with deionized water, 0.01 mol/L HCl, and 0.01 mol/L NaOH for 0.5 h each. Following the treatment, the membrane was carefully removed, and both sides of the membrane were thoroughly rinsed multiple times with deionized water. To evaluate the antifouling performance of the regenerated cellulose membrane, the membrane flux recovery rate (r) was determined by comparing the water flux before and after cleaning. The calculation formula is the same as mentioned in Section 1.

## 3. Results and Discussion

### 3.1. Raw Water Quality for Drinking Water

Turbidity is a vital indicator when assessing the quality of drinking water. When water contains impurities such as soil, silt, fine organic and inorganic matter, plankton, and other suspended particles, these impurities adhere to the water’s surface, resulting in turbidity and the presence of certain turbidity levels. This serves as a proxy parameter for suspended matter, reflecting the quantity of impurities in the water. According to the “Sanitary Standards for Drinking Water,” turbidity should not exceed 1 NTU. Therefore, reducing turbidity levels in drinking water is crucial for improving water quality.

The turbidity of raw drinking water is highly susceptible to changes in climatic conditions. Therefore, to ensure water quality accuracy, water samples were taken every two days during autumn, and the turbidity of these samples was measured using a portable turbidity meter and recorded. By monitoring changes in turbidity, water quality conditions were evaluated and the quarterly average turbidity was calculated to assess water quality changes before and after membrane treatment.

As illustrated in Figure 2, both the turbidity and chromaticity of water were highest during the first few days of detection. Apart from summer rainfall, corrosion cannot be ruled out as a contributing factor, resulting in a yellowish tint that is visible to the naked eye. The water samples were intermittently stirred during measurement to minimize the effects of other factors on the readings. Due to the influence of autumn rainfall and temperature, rainwater mixed with sediment had a greater impact on water quality, resulting in an increase in water sample color that exceeded the detection indicators established by the “drinking water health standards.” This highlights the importance of accounting for these environmental factors when evaluating drinking water purification performance.

Chemical oxygen demand (COD) is often used as an indicator to measure the concentration of organic matter in water. According to the test indices outlined in the “Sanitary Standards for Drinking Water” (GB5749-2022), the COD_MN_ and ammonia nitrogen values of water samples should not exceed 3 mg/L and 0.5 mg/L, respectively. Figure 3 illustrates the changes in COD_MN_ and ammonia nitrogen values of water samples during the experiment.

After 30 days of sampling and detection, the measured values of ammonia nitrogen were mostly below 0.5 mg/L, which met the detection standards. It was also noted that water turbidity tended to increase when the COD_MN_ value was high, indicating that the organic matter content was related to the turbidity. However, the measured value of COD_MN_ fluctuated at approximately 3 mg/L, indicating that the raw water quality had a high content of organic matter, meaning that the water required purification in order to be safe for human consumption.

### 3.2. Ultrafiltration + Nanofiltration Combined Membrane Process Treatment

The selection of ultrafiltration and nanofiltration process parameters for purifying drinking water in an industrial context prioritized cost-effectiveness, with a preference for room temperature operation to minimize energy consumption and equipment costs, ensuring overall economic feasibility. In the UF stage, a pressure of 0.1 MPa was chosen to balance energy consumption and membrane permeability. For the NF stage, a pressure of 0.5 MPa was selected to more effectively remove minute particles and dissolved substances.

Therefore, the membrane combination process was conducted at 25 °C, with ultrafiltration at 0.1 MPa and nanofiltration at 0.5 MPa. The results of the ultrafiltration–nanofiltration membrane combination treatment on water quality are presented in Table 1. As summarized in the table, the combination of ultrafiltration and nanofiltration membranes had a significant effect on turbidity treatment, with the turbidity of water quality being stabilized at a level below 0.1 NTU. This suggests that the ultrafiltration and nanofiltration process effectively intercepted waterborne contaminants, ensuring the microbial safety of drinking water and meeting high-quality drinking water standards. When the turbidity decreased below 0.1 NTU, particles in the water were undetectable, and the color of the water before and after membrane treatment decreased to 2 degrees, indicating a significant sensory difference.

The change in pH before and after the experimental treatment was minimal, and it remained below the standard limit for drinking water quality. The measured COD_MN_ value of raw water quality fluctuated around the standard value, and the value decreased to 0.8095 mg/L after the ultrafiltration–nanofiltration membrane combination treatment, with a removal rate of 75%. The ultrafiltration and nanofiltration membrane combination also demonstrated excellent treatment effect for ammonia nitrogen, with a removal rate of 88.6%. Total water hardness is associated with the taste of drinking water, and the total hardness of raw water in the experiment was below the standard value both before and after the ultrafiltration and nanofiltration membrane combination process, meeting the requirements for high-quality drinking water.

To ensure the accuracy of the ultrafiltration and nanofiltration membrane combination process for drinking water purification, the effects of turbidity, COD_MN_, and ammonia nitrogen treatments were analyzed by increasing the number of experiments. Additionally, by comparing the water quality of drinking water treated by the ultrafiltration unit and “ultrafiltration + nanofiltration,” the treatment effects of each filter unit in the combined ultrafiltration and nanofiltration membrane process for drinking water were explored.

As shown in Figure 4, the ultrafiltration membrane has a stable turbidity removal effect of over 80% at a range of 0.10–0.15 NTU. This efficacy is attributed to the ultrafiltration’s capacity to adsorb and mechanically trap suspended turbidity in drinking water. However, some small pollutants still manage to pass through the pores of the ultrafiltration membrane, making complete removal challenging. The double-membrane process achieves a turbidity range of 0.08–0.03 NTU, with a stable turbidity removal rate of over 95%. The addition of a nanofiltration membrane in the combined membrane process enhances the mechanical retention ability of pollutants in water, showcasing excellent performance in reducing water turbidity. This demonstrates that the experimental ZrO_2_/BCM and IP-ZrO_2_/BCM-NFM combined “ultrafiltration + nanofiltration” cellulose membrane assembly technology meets the turbidity purification requirements.

COD_MN_, also known as oxygen consumption, is an indicator of organic matter pollution in water. Figure 5 shows the measured values of COD_MN_ after treatment by each filtration unit in the ultrafilter–nanofiltration membrane combination process. The ultrafiltration unit has a poor treatment effect on COD_MN_, with a stable removal rate of about 20–25%. This is because COD_MN_ is partly composed of suspended organic matter, colloidal state, and soluble organic matter, while the interception and screening of the ultrafiltration unit are only for suspended organic matter. Moreover, suspended organic matter is also related to turbidity, which explains the good removal effect of the ultrafiltration unit on turbidity.

Therefore, in the combined ultrafiltration and nanofiltration membrane process, the nanofiltration unit plays a greater role in removing organic matter. The nanofiltration membrane can maintain a COD_MN_ removal rate above 70% due to its low-molecular-weight cut-off and charge adsorption characteristics on the membrane surface. The final effluent quality of the ultrafiltration and nanofiltration membrane combination process has a COD_MN_ value below 1 mg/L, indicating excellent effluent quality.

As shown in Figure 6, the technology of combined ultrafiltration and nanofiltration membrane is used for ammonia nitrogen removal in water. The ultrafiltration unit has a low efficiency in removing ammonia nitrogen, with a removal rate of about 10%. The overall removal rate is only between 63% and 72%. This is because the molecular weight of ammonia nitrogen is very low, almost the same as that of water. Without other process conditions such as aeration, only the pore size and negative charge of the nanofiltration membrane can intercept ammonia nitrogen. The measured value of ammonia nitrogen after each process of combined ultrafiltration and nanofiltration membrane was about 0.1 mg/L, which meets excellent drinking water standards.

The comparison of pollutant removal efficiency between a single ultrafiltration membrane and the combined membrane process falls short of conclusively demonstrating the superiority of the combined membrane process over standalone ultrafiltration. In order to conduct a more comprehensive evaluation, traditional nanofiltration membranes were introduced in the study, allowing for a comparative analysis of their performance in pollutant removal against that of the membrane combination process. This approach is designed to accurately illustrate the enhanced capabilities of the membrane combination process in pollutant removal from water, particularly when compared to traditional nanofiltration membranes.

As shown in Figure 2, Figure 3, Figure 4, Figure 5, Figure 6 and Figure 7, a single traditional nanofiltration membrane achieved a turbidity removal ranging from 81% to 93%. In comparison, the membrane combination process exhibited a turbidity removal rate between 91% and 95%. The membrane combination process not only undergoes the first purification of water through ultrafiltration, filtering out the majority of suspended solids, but also performs a secondary purification through nanofiltration, more thoroughly removing suspended solids from the water, resulting in a decrease in turbidity. For a single traditional nanofiltration membrane, the removal rate of COD_MN_ falls between 62% and 71%, while the membrane combination process achieves a COD_MN_ removal rate of 70% to 79%. Since the membrane combination process involves filtration through both ultrafiltration and nanofiltration membranes, during the ultrafiltration stage, organic substances in the water react with the ZrO_2_ on the surface of the ultrafiltration membrane, reducing a portion of them. Therefore, compared to a single traditional nanofiltration membrane for COD_MN_, the membrane combination process achieves a more thorough removal. The removal rate of ammonia nitrogen for a single traditional nanofiltration membrane ranges from 61% to 65%, while the membrane combination process can achieve a maximum of 70%. This clearly indicates that the adoption of membrane combination technology for drinking water purification is far superior to the performance of a single traditional nanofiltration membrane.

Krystyna Konieczny et al. employed traditional nanofiltration and ultrafiltration membrane combination processes for wastewater treatment [44]. However, the treated water did not meet the standards for drinking water. In contrast, in this study, the combination membranes ZrO_2_/BCM and IP-ZrO_2_/BC-NFM were used for water treatment, and the treated water met all the standards for drinking water. This was attributed to the addition of ZrO_2_ to the membrane, enhancing the mechanical retention rate of the ultrafiltration membrane and the pollutant retention rate of the nanofiltration membrane compared to the traditional membrane.

### 3.3. The Performance of Membrane Components

The fouling of membrane components is a crucial factor that limits the performance of membrane treatment. Membrane fouling is mainly caused by protein and humic acid deposition. In this study, the bovine serum protein interception by ZrO_2_/BCM was tested to evaluate the contamination resistance of the cellulose ultrafiltration membrane. The separation performance of the IP-ZrO_2_/BCM-NFM nanofiltration membrane was tested by measuring the retention of dissolved inorganic salts. The experiments in this chapter, respectively, tested the retention effect of Na_2_SO_4_ and BSA and used Na_2_SO_4_ + BSA to simulate pollutants in water. The anti-fouling performance of the nanofiltration unit was tested, and the long-term operation was used to evaluate the anti-fouling performance and stability of the membrane by maintaining the pollutant rejection rate.

To evaluate the stability of the nanofiltration unit, long-term continuous filtration tests were performed on the membrane at room temperature and under a pressure of 0.5 MPa, as shown in Figure 2, Figure 3, Figure 4, Figure 5, Figure 6, Figure 7 and Figure 8, to assess its ability to intercept pollutants. The rejection rates of Na_2_SO_4_ and BSA by the nanofiltration unit remained stable, with a change range of less than 5%. The IP-ZrO_2_/BCM-NFM membrane showed excellent performance in separating divalent salt solution from bovine serum protein. Li et al.’s single cellulose nanofiltration membrane exhibited a retention rate of only 71.23% for Na_2_SO_4_ [32]. In comparison, the utilization of combined membrane technology demonstrates a more pronounced impact on inorganic salt retention than that observed with a single nanofiltration membrane. This enhanced performance is attributed to the synergistic sieving effect and charge repulsion resulting from the pore sizes of both ultrafiltration and nanofiltration membranes. Despite the fact that the addition of ZrO_2_ particles in a single ZrO_2_/BCM ultrafiltration membrane aids in filling pores, improving membrane resistance, and achieving a BSA retention rate exceeding 91% [42], the retention rate of the combined membrane technology surpasses that of the single ZrO_2_/BCM, reaching an impressive 99.99%. This superiority is ascribed to the smaller pore size of the nanofiltration membrane in the composite structure, significantly amplifying the sieving effect of the pore size and enabling the interception of macromolecular pollutants passing through the ultrafiltration membrane.

As illustrated in Figure 9, the membrane flux of the nanofiltration unit showed a gradual decrease when Na_2_SO_4_ + BSA was used as the pollutant. The decline in water flux was attributed to membrane fouling caused by the adhesion of BSA on the membrane surface, which further aggravated the membrane pore blockage of the nanofiltration unit with the increase in operation time. Once the adsorption and desorption of pollutants reached an equilibrium state, the membrane flux tended to be stable with a decrease rate of approximately 25%. Zheng Kai discovered, after running for 36 h, the BSA membrane flux of the MgO blended polyamide composite nanofiltration membrane decreased by 35% [45]. Conventional polyamide nanofiltration membranes take a long time to achieve stable water flux, whereas in this study, the water flux of IP-ZrO_2_/BCM-NFM tended to be stable after 9 h of test operation. Due to the membrane combination process, the majority of macromolecular substances and inorganic salts are mechanically intercepted by the ultrafiltration membrane before the raw water transits through the nanofiltration membrane. This meticulous process substantially mitigates pollution and blockage on the nanofiltration membrane’s surface, thereby upholding a consistent and stable water flux. This indicates that the nanofiltration unit can maintain stable performance under long-term operation.

### 3.4. The Cleaning of Membranes

During the continuous operation of the “ultrafiltration + nanofiltration” membrane combination process, a certain fluid shear force is exerted on the membrane surface, which can effectively mitigate fouling on the membrane surface. However, the pollutants that foul the membrane surface tend to gradually accumulate over time, ultimately resulting in irreversible changes. Therefore, the membrane flux was tested before and after contamination to evaluate the separation performance and long-term stability of the membrane combination.

In this study, the contaminated cellulose membrane was subjected to cleaning with water, acid, and alkali, respectively, using the membrane filtration system. Figure 2, Figure 3, Figure 4, Figure 5, Figure 6, Figure 7, Figure 8, Figure 9 and Figure 10 show the membrane flux recovery rate after cleaning the cellulose membrane with the same pollutant interception under different conditions. It is observed that when water is used as the cleaning agent, the recovery rate of membrane flux is significantly lower than that of acid and alkali washing. The flux of modified ZrO_2_/BCM after washing can be recovered to 86%, which indicates that the addition of ZrO_2_ effectively reduces the contact between pollutants and the membrane surface. The membrane flux recovery rate of IP-ZrO_2_/BC-NFM is slightly lower than that of ZrO_2_/BCM, which may be due to the smaller pore size of the nanofiltration membrane. The nanofiltration membrane can effectively intercept pollutants, while the pollutants stay in a smaller pore size, which is difficult to remove.

After cleaning with HCl and NaOH, the flux recovery rate of the membrane reached more than 90%. Some related studies also obtained the result of flux improvement by chemical cleaning of cellulose, but studies showed that the integrity of cellulose membrane was vulnerable to damage under the condition of strong acid and alkali, leading to a high flux recovery rate [46]. Similarly, for IP-ZrO_2_/BC-NFM, acidic and alkaline cleaning may change the properties of the PA layer of the nanofiltration membrane, protonating the N or O atoms of the amide group, thereby reducing the stability of the polyamide layer [47]. However, in this study, IP-ZrO_2_/BC-NFM showed good acid resistance, so it could maintain the stable performance of the membrane structure while removing pollutants by chemical cleaning.

## 4. Conclusions

The “ultrafiltration + nanofiltration” cellulose membrane assembly technology combining ZrO_2_/BCM and IP-ZrO_2_/BCM-NFM, as explored in this chapter, was confirmed to be effective in the comprehensive treatment of raw water. The integration of ultrafiltration and nanofiltration membranes remarkably enhanced turbidity reduction, consistently maintaining the turbidity of the membrane-filtered water at below 0.1 NTU. Furthermore, this treatment approach yielded removal rates of 75% for COD_MN_ and an impressive 88.6% for ammonia nitrogen, thus effectively meeting the stringent standards for the production of high-quality drinking water. It reduces the probability of people getting sick from drinking water that does not meet drinking water standards.

Our experimental findings underscore the exceptional performance of the “ultrafiltration + nanofiltration” cellulose membrane combination. Turbidity removal consistently exceeded 95%, and COD_MN_ removal rates remained consistently above 70%. Notably, the nanofiltration unit exhibited remarkable stability in rejecting Na_2_SO_4_ and BSA, with deviations of less than 5% after 3 h of continuous testing. Even during an extended 9 h operation period, the decline in water flux from the nanofiltration unit was limited to approximately 25%. These results demonstrate the capacity of the “ultrafiltration + nanofiltration” cellulose membrane combination to maintain robust and reliable performance over prolonged operational periods. Collectively, our findings highlight the potential of our proposed membrane assembly technology as a sustainable and effective solution for the treatment of raw water, offering a viable pathway for the purification of water to meet the highest standards of quality, especially in the context of producing safe and clean drinking water.

## Figures and Tables

**Figure 1 polymers-16-00285-f001:**
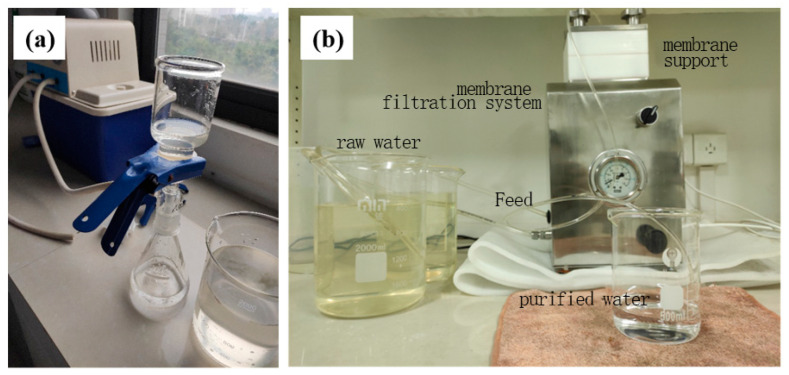
“Ultrafiltration + Nanofiltration” cellulose membrane component treatment process: (**a**) microfiltration membrane filtration; (**b**) membrane filtration system.

**Figure 2 polymers-16-00285-f002:**
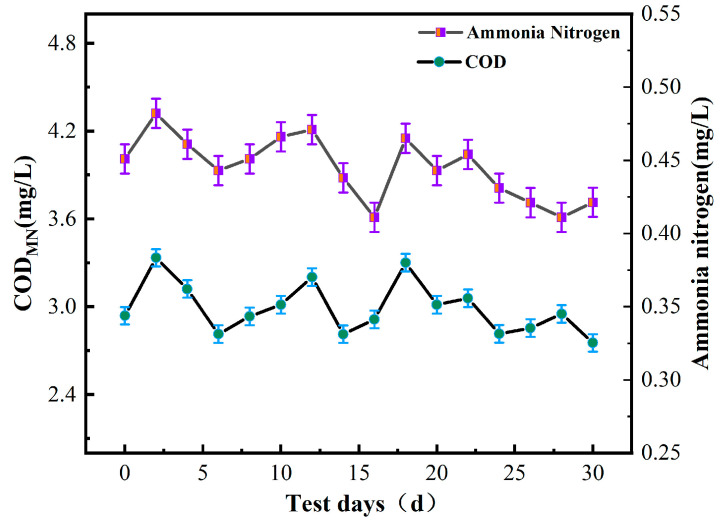
Turbidity and chromaticity in water.

**Figure 3 polymers-16-00285-f003:**
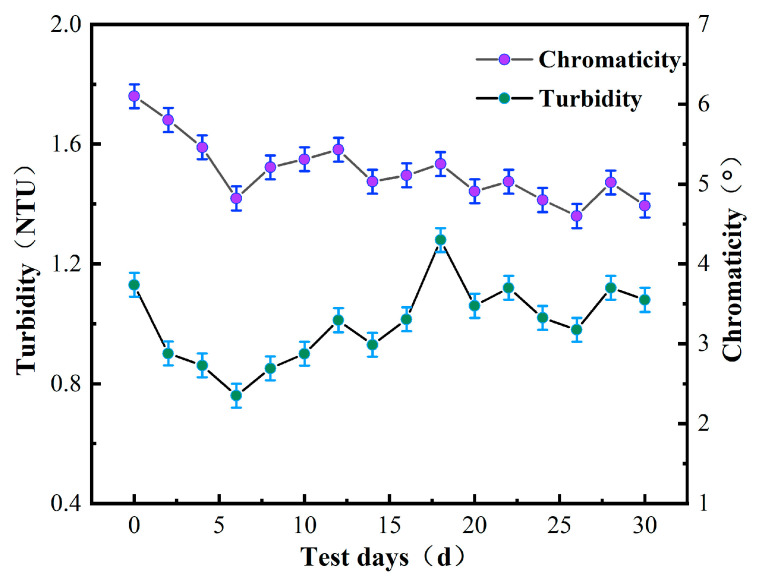
Ammonia nitrogen and COD_MN_ in water.

**Figure 4 polymers-16-00285-f004:**
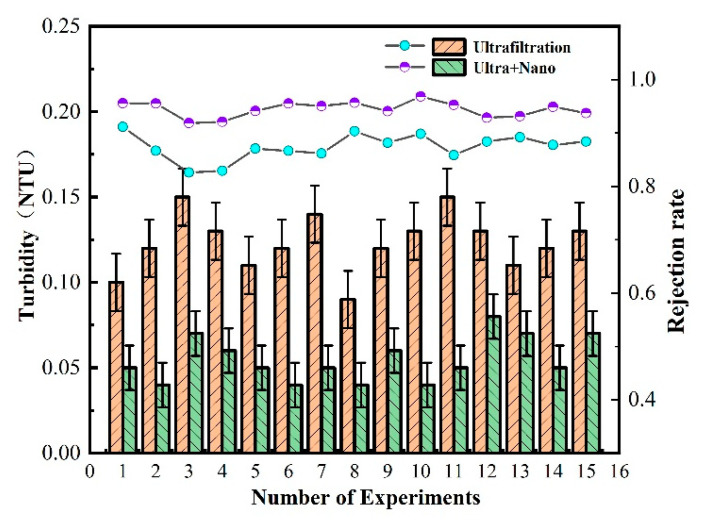
Removal efficiency of membrane combination process for turbidity in water quality.

**Figure 5 polymers-16-00285-f005:**
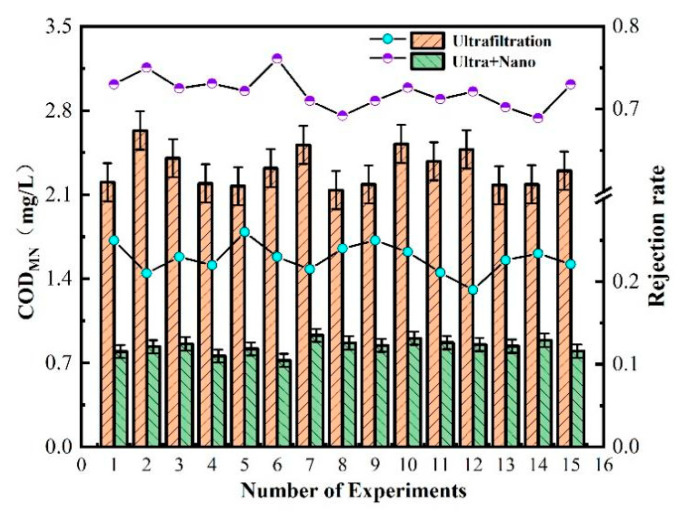
Removal efficiency of membrane combination process for COD_MN_ in water quality.

**Figure 6 polymers-16-00285-f006:**
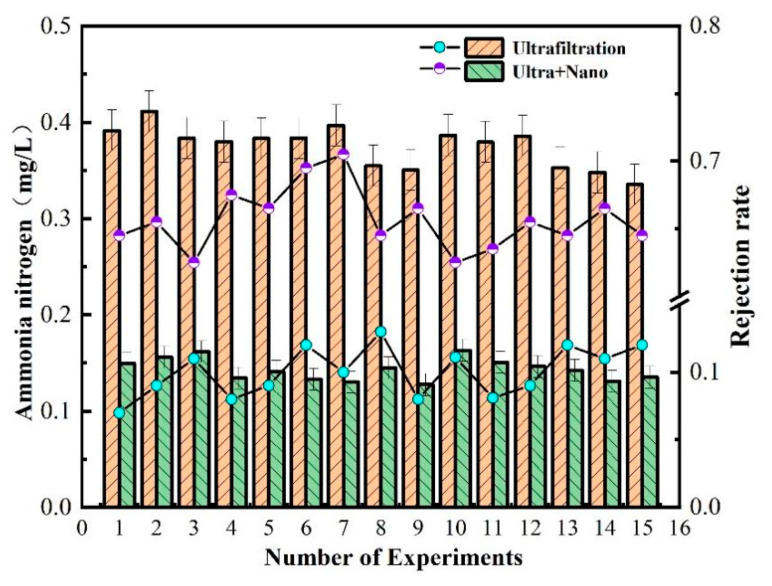
Removal efficiency of ammonia nitrogen in water by membrane combination process.

**Figure 7 polymers-16-00285-f007:**
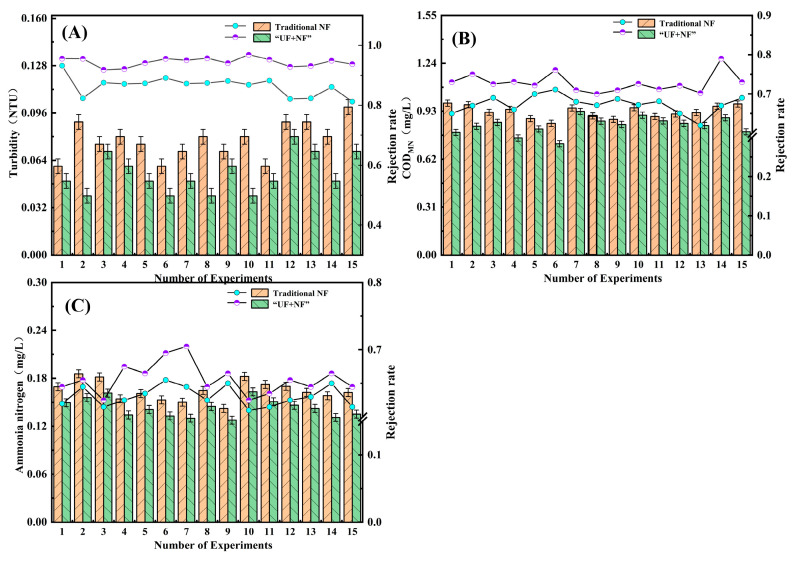
Comparison of turbidity, COD_MN_ and ammonia nitrogen removal between conventional nanofiltration membrane and combined membrane process: (**A**) removal efficiency for turbidity in water quality; (**B**) removal efficiency for COD_MN_ in water quality; (**C**) Removal efficiency for ammonia nitrogen in water.

**Figure 8 polymers-16-00285-f008:**
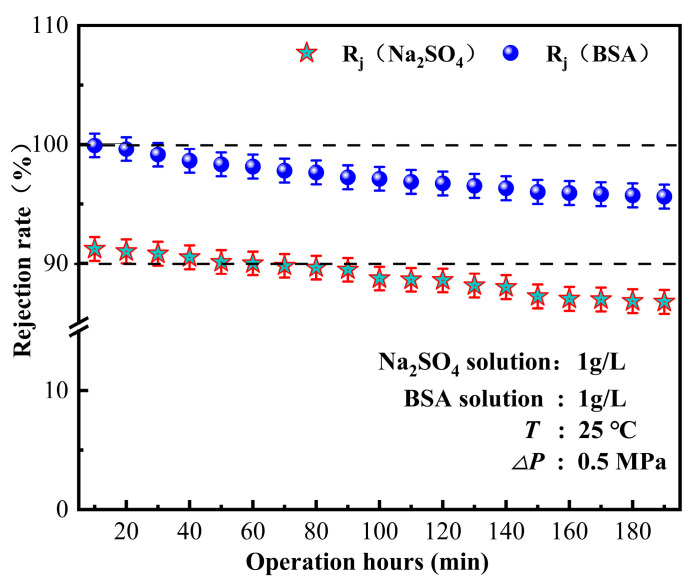
The retention effect of pollutants by nanofiltration unit.

**Figure 9 polymers-16-00285-f009:**
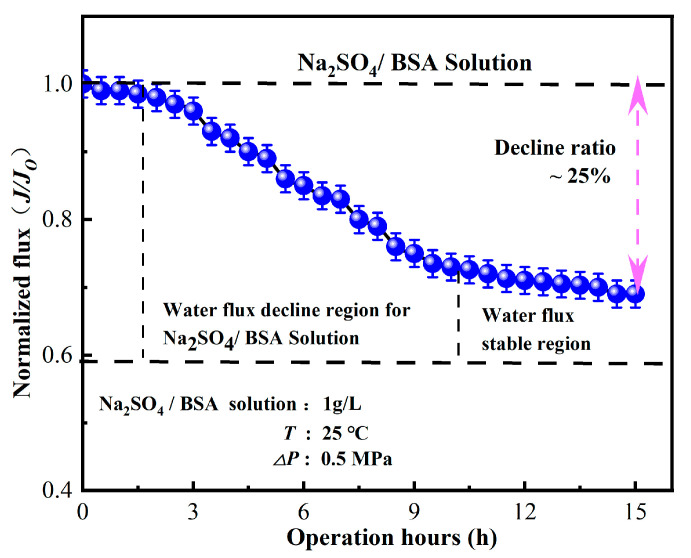
Influence of operating time on separation performance of nanofiltration unit.

**Figure 10 polymers-16-00285-f010:**
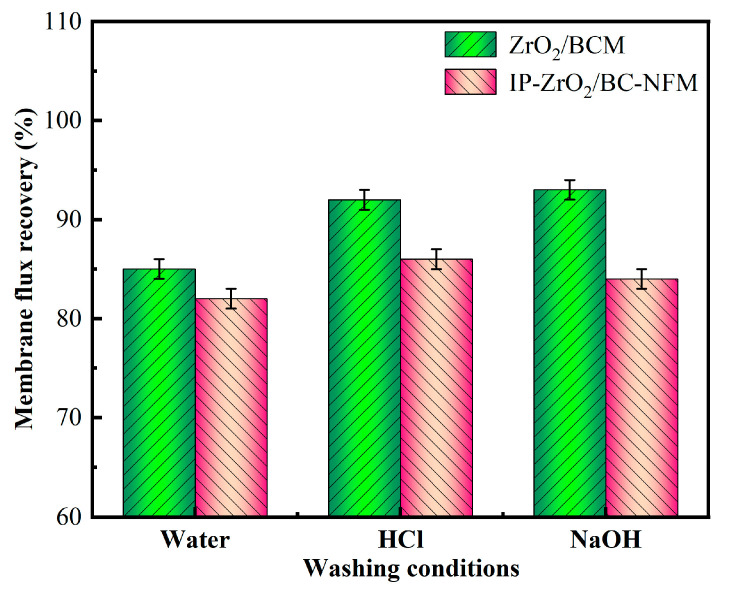
Membrane flux recovery under different cleaning conditions.

**Table 1 polymers-16-00285-t001:** Water quality detection after ultrafiltration–nanofiltration membrane combined treatment.

Water Quality Index	Tap Water Quality	Filtered Water Quality	Sanitary Standards for Drinking Water Quality Standards	Standard for Drinking Water Purification
Chromacity	6 ± 0.1	2 ± 0.1	≤5	
Turbidity (NTU)	1.03 ± 0.01	0.08 ± 0.01	≤1.0	≤0.5
COD_MN_ (mg/L)	3.238 ± 0.01	0.8095 ± 0.01	≤3.0	≤2.0
pH	7.54 ± 0.02	7.29 ± 0.02	6.5~8.5	6.5~8.5
Ammonia Nitrogen (mg/L)	0.411 ± 0.002	0.097 ± 0.002	≤0.5	≤0.5
Total Hardness (mg/L)	184.3 ± 1.0	60.7 ± 1.0	≤450	≤300

## Data Availability

All relevant data are included in the paper.

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
