# Peer review of "The Performance of Cellulose Composite Membranes and Their Application in Drinking Water Treatment"

_polymers, 2024, doi:10.3390/polym16020285_

Round 1

Reviewer 1 Report

Comments and Suggestions for Authors

Development and new and efficient methods for drinking water purification has high relevance for the practice. Membrane separation processes can answer for this challenge and these methods are suitable to remove the micropollutants as well. Fabrication of new membrane using new membrane materials and surface modifications are considered as promising methods to achieve high permeate flux and good fouling behaviour with high pollutants retention efficiency. The manuscript has a special focus on the applicability and efficiency of cellulose composite (ZrO2/BCM) ultrafiltration membranes and IP-ZrO2/BC NFM nanofiltration membranes. The manuscript contains interesting data and information, but results are not discussed in details and methodological information are missing (see my comments), respectively.

Some specific comments:

Authors did not investigate the efficiency of one-step NF process (compared to the two stages UF-NF).

There are known ’conventional’ processes for COD, turbidity and nitrogen removal. it should be tested the efficiency of these process vs. UF-NF.

The structural/morphological parameters of the fabricated membranes are not investigated.

The experimental results are not compared and discussed with references.

UF but mainly NF need high pressure, therefore the costs of the combined process should be analysed, as well.

In the Abstract should be given the operational parameters of UF and NF stages, as well.

In lines 36, 41 and 45 please discuss briefly the main findings of the given (numerous) references.

Please provide briefly the membrane fabrication methods (condition, parameters).

Please give information how the UF and NF process parameters (pressure, temperature etc) are selected/determined.

Measuring errors/standard deviation of data presented in Figure2, 3, 7, 8 and Table 1 are not presented.

The citation and reference style in the MS text (see line 347, 350, for instances) and reference list is not unified.

Reviewer 2 Report

Comments and Suggestions for Authors

The manuscript “Performance of Cellulose Composite Membranes and Their Application in Drinking Water Treatment” presents an interesting solution to the water purification. The manuscript has been written in good grammar and syntax. The manuscript may be considered for publication after response to the following comments. My comments are as follows:

1.       The study produced two membranes ZrO2/BCM and  IP-ZrO2/BC-NFM. What is the purpose of producing two membranes? These membranes alone are sufficient for water treatment.

2.       The data presented in the manuscript is for the combined membrane. What about the efficiency of the individual membranes?

3.       Although the membrane is efficient in drinking water purification. What specific components can be removed from wastewater using this method? Is there a specific wastewater or industry that may be targeted?

4.       Line 26: “treatment of drinking water,”. Is it going to treat drinking water? Or wastewater?

5.       At end of the abstract and conclusion a note should be added for the specificity of the membrane for removal of a particular contaminant from the wastewater based on the results presented in the study.

6.       The discussion of the manuscript is too weak. The manuscript lacks a discussion and comparison with the previous report in result and discussion.

7.       The manuscript should be improved by comparing and discussing previous findings from the literature.

8.       Figure legends such as” Figure 2. Turbidity and chromaticity in water.” Lacks information. All the figure legend should at least provide the presentation and major components so that it is stand alone.

9.       The study need to be compared with recent findings related to the subject area.

Round 2

Reviewer 1 Report

Comments and Suggestions for Authors

The authors have revised the mnauscript thoroughly according to reviewers' comments and suggestions. Rephrasings, more detailed Introduction and Discussion , and more detailed methodology parts made the manuscript more complete and clearer. The overall scientific quality of the mnauscript has been improved significantly.

Comments:

Please use the measured parameter or the rejection rate in the y axis in Figure 7 (please unify).

Please provide standard deviation/measuring erreors for flux recovery in Figure 10.

Please check and unify the reference style (see line 411, for instance).

Reviewer 2 Report

Comments and Suggestions for Authors

The manuscript is sufficiently revised. 
